# Determinants of the Underlying Causes of Mortality during the First Wave of COVID-19 Pandemic in Saudi Arabia: A Descriptive National Study

**DOI:** 10.3390/ijerph182312647

**Published:** 2021-11-30

**Authors:** Abdullah M. Asiri, Shaker A. Alomary, Saeed A. Alqahtani, Izzeldin F. Adam, Samar A. Amer

**Affiliations:** 1Department of Infectious Diseases, Preventive Health, Ministry of Health, Riyadh 11176, Saudi Arabia; abdullahm.asiri@moh.gov.sa; 2Department of Health Programs and Chronic Diseases, Ministry of Health, Riyadh 11176, Saudi Arabia; Shalomary@moh.gov.sa (S.A.A.); Sahalqahtani@moh.gov.sa (S.A.A.); iabdellah@moh.gov.sa (I.F.A.); 3Department of Epidemiology, University of Khartoum, Khartoum 11115, Sudan; 4Department of Public Health and Community Medicine, Zagazig University, Zagazig 44519, Egypt

**Keywords:** COVID-19, international classification of diseases, Kingdom of Saudi Arabia, underlying mortality

## Abstract

Since the emergence of the COVID-19 pandemic, the mortality statistics are constantly changing globally. Mortality statistics analysis has vital implications to implement evidence-based policy recommendations. This study aims to study the demographic characteristics, patterns, determinants, and the main causes of death during the first half of 2020, in the Kingdom of Saudi Arabia (KSA). Methodology: A retrospective descriptive study targeted all death (29,291) registered in 286 private and governmental health settings, from all over KSA. The data was extracted from the ministry of health’s death records after the ethical approval. The International Classification of Diseases (ICD-10) and WHO grouping, were used to classify the underlying causes of deaths. The collected data were analyzed using the appropriate tables and graphs. Results: 7055 (24.9%) died at the middle age (40–59 year), and 19,212 (65.6%) were males, and 18,110 (61.8%) were Saudi. The leading causes of deaths were non-communicable diseases (NCDs) 15,340 (62.1%), mainly Cardiovascular diseases (CVDs) 10,103 (34.5%). There was a significant relationship between the main causes of deaths and sex (*p* < 0.05) and nationality (*p* = 0.01). Conclusion: NCDs mainly CVDs are the leading cause of death. The COVID-19 mortalities were mainly in males, and old age > 55 year. The lockdown was associated with a reduction in the NCDs and Road traffic accidents mortalities.

## 1. Introduction

World Health Organization (WHO), On 19 March 2020, declared COVID-19 as a pandemic [1]. It becomes one of the deadliest pandemics in the last century [2]. Worldwide, On, 20 July 2020, the deaths have already exceeded 609,000. While in the Kingdome of Saudi Arabia (KSA), there were 2486 deaths to be on the 27th rank according to the number of deaths [3].

COVID-19 pandemic affects the entire health system resulting in potentially unexpected worse health outcomes and increases the overall mortality [4,5]. The interaction between COVID-19 and NCDs may increase the global burden of disease (GBD) [6].

Globally, NCDs, are responsible for two out of every three deaths [7] and from all deaths accounted for greater than 70% in 2017 [8] and approximately 73% with more than 90,000 deaths per year in KSA [9]. NCDs deaths are mainly due to cardiovascular diseases (CVDs), cancers, chronic respiratory diseases, and diabetes [10,11,12]; other major causes include suicide and injury [13].

Mortality data derived from death certificates are used to measure the population health and to compare mortalities at the national and international levels [14]. It reflects the social, economic, environmental conditions and the provided health care [15,16].

The determinants of mortality facilitate the planning and evaluation of health policies and programs. The mortality trends highlight the urgency of adapting health systems to better meet the changing healthcare needs, by improving the capacity to prevent and manage NCDs [6] because the disease progress is not uniform continuously changes [16]. Therefore, the reported trends can be incorporated into the adaptation process.

Mortality indices should be a part of the country system backbone continuously, as important health care quality markers, provide essential inputs for national and subnational health policy and planning [7], and help in predicting and evaluating many disease prevention initiatives and health promotion programs [17].

The scientific information available on the status and trends of various Communicable Diseases (CDs), NCDs, and deaths are limited to the primary statistics available in the portals of the Ministry of Health (MOH) and other health agencies [18]. However, studies on these available data in KSA are scanty and scattered. Although KSA is the largest Arab state in Western Asia and is part of the “Group of Twenty” (G-20) of major economies [19], it still requires accurate and comprehensive relevant national health data.

Hence, the present study aimed; to provide relevant national-based mortality statistics to guide the government and policymakers for the optimal resources allocation to enhancing life expectancy and reducing mortality in KSA. The study attempted to determine the demographic characteristic, patterns, determinants, and underlying causes of mortality during the first half of 2020 during the first wave of COVID-19 pandemic in KSA.

## 2. Methodology

### 2.1. Participants

A retrospective descriptive study, from 1 January until the end of June 2020. Record based study targeted all the registered mortalities from 286 health settings {governmental hospitals (85.0%), medical cities (5.0%), governmental health centers (5.0%), private hospitals (4.0%) and specialized governmental hospitals (1.0%)} all over the 20 health regions in KSA.

### 2.2. Data Collection

Extracted data about three main parts: demographic data (age, sex, and nationality), mortality data include (date of death, the underlying cause of death coded using the ICD-10), and the regions.

The causes of death are classified according to The International Classification of Diseases (ICD-10) [20] and WHO classification [21] We reassigned deaths with ill-defined or codes to be allocated to other causes using previously published methods as described by Mathers et al. [22], e.g., cancers of the unspecified site to cancer, ill-defined cardiovascular causes of death to cardiovascular causes of death. Finally, we redistributed ill-defined causes of death data to all non-injury causes of death.

### 2.3. Statistical Analysis

The collected data was coded and analyzed using SPSS (version 22). At a level of significance (*p*-value ≤ 0.05) and calculating the following rates. Mortality Rates were calculated using the WHO reference population [21] and the Mortality was projected using methods applied in other studies [23,24].

## 3. Results

Out of the 31,509 deaths in KSA during the first half of 2020, the causes of death were available for 29,291 (92%). They were 7055 (24.9%) were middle-aged (40–59 year), and 19,212 (65.6%) were males, and 18,110 (61.8%) were Citizens.

According to the ICD-10 classification, the underlying categories of morality in descending order were 6864 (23.4%) CVDs, 2773 (9.4%) ill-defined injury conditions, 2438 (8.3%) infectious and parasitic diseases, 1421 (4.9%) injuries and poisoning, 1199 (4.1%) cancer, 1153 (3.9%) chronic respiratory disorders, 994 (3.4%) RTTA, 864 (2.9%) Acute Respiratory Infections (ARIs), 658 (2.2%) and diabetes, 640 (2.2) (Table 1).

We found that the main CDs underlying causes of mortalities based on ICD-10 (20) were 3317 (11.3%), out of them 167 (5.0) Streptococcal infection, 95 (2.9%) Varicella 90 (2.7%), 78(2.4%) Mycoplasma Pneumonia, 56 (1. 7%) Tuberculosis (TB), and 42 (1.3%) Hepatitis.

Table 2 shows that according to ICD-10 and WHO grouping the top leading underlying causes of mortalities in descending orders were NCDs 15,340 (62.1%) including CVDs 10,103 (34.5%), {Communicable, maternal, and perinatal 5546 (18.9)}, Ill-defined causes 4060(13.9%), and Injuries 2860 (9.8%).

The national lockdown precautions across months as described in detail as illustrated in (Figure 1), the pattern and the total number of mortalities including mortalities of NCDs and RTTA decreased but COVID-19 deaths progressively increased (Figure 2).

Analyzing the trends of the main causes of death across months reveals a major shift in the main causes of deaths. As CDs showed a progressive increase from 617 (10.8%) at the end of January to 717 (18.5%) at the end of April, to be 1620 (26.1%) at the end of June. While RTTA and NCDs showed a manifest decrease from 238 (4.1%), 3019 (53.2%) reciprocally at the end of January to be 114 (1.9%), and 2146 (55.4%) at the end of April then gradually increased to be 147 (2.4%), and 2894 (47.0%) at the end of June. respectively depicting how drastic the percentages and burden have changed (Figure 2).

COVID-19 age-specific mortality rate was less than 2/100,000 population at the age groups <35 year, and progressively increased to be 17.7/100,000 at the age group (50–54 year) then progressively increases till reaching the peak 74.9/100,000 at the age 80 year and more. While CDs other than COVID-19 mortality was 8.4/100,000 at the age group 0–4 year then stated a progressive increase from 17.7/10,000 at the age of 55–59 year until reach the highest of 529.5/100,000 at the age 80 year and more (Table 3).

The top regions regarding NCDs mortalities were Al-Ahasa, Jeddah, Riyadh, and Makkah, by 553 (70.5%), 2620 (17.1 %), 2021 (13.2%), and 1936 (12.6%). The lowest was in Al-Qurayyat 92 (0.6%). While the top regions regarding RTTA mortalities are Riyadh, Makkah, Jeddah, and Asser by 23.3%, 9.3%, 7.8%, and 7.4% reciprocal and the lowest was in Al Jouf (0.4%) (Table 4).

In KSA, 19,212 (65.6%) of mortalities were males but the SSMR was nearly equal in both sexes. NCDs were the main cause of mortalities in both sexes. There was a statistically significant relationship between the sex and nationalities and main causes of deaths as the Proportional mortalities per sex from COVID-19 more in males 1361(7.1%) but from NCDs more in females 5916 (58.7%) (Table 5 and Table 6).

## 4. Discussion

Mortality statistics are important measures during the COVID-19 pandemic [24] Comparable information about deaths and mortality rates as a starting point for informed health policy debate is essential. GBD follows the principles of ICD that a single underlying cause was assigned to each death [25].

Despite efforts to improve the quality and availability of mortality statistics, they are still, in a deplorable state as 4060 (13.9%) of mortalities are classified as ill-defined causes and 1485 (5.0%) as a null cause. This may be due to the urgent circumstances during the pandemic, as there was no access to standard care [26]. Similar problems are reported in disease reporting systems globally, e.g., weak infrastructure and rules [27,28,29]. From the total mortalities, the cause of death is assigned for no more than one-third and often considerably uncertain diagnosis [30].

Regarding NCDs mortality, NCDs encompass a vast group of diseases [31]. In the underlying NCDs causes of deaths in descending order were CVDs 10,103 (34.5%), cancer 1345 (4.6%), chronic respiratory diseases 1053 (3.9%), and diabetes 658 (2.2%). Inconsistent with the annual burden of global NCDs mortalities as CVDs diseases account for (17.7 million), followed by cancers (8.8 million), respiratory diseases (3.9 million), and diabetes (1.6 million) [32].

In this study, NCDs responsible for 15,340 (62.1%) of all mortalities, 13,159 (85.3%) of them are caused by the four disease clusters (CVDs, cancers, chronic respiratory diseases, and diabetes), while worldwide NCDs kill nearly 40 million people a year equivalent to 70% of all deaths, 32 million (80%) of all NCDs deaths are caused by the four disease clusters [32]. Therefore, the burden of NCDs is rising rapidly and has now become a major challenge to global development. This rapid increasing is due to multiple risk factors such as physical inactivity, sedentary lifestyle, economic development, increase in the use of tobacco, increase in the intake of processed foods high in sugar, fat, and salt, and change in cultural norms [33].

Comparing our results to the WHO country profile report in 2016, the overall mortalities from NCDs CVDs, and cancer are decreased from (73% to 62.1%), (37% to 34.6) and (10% to 4.6%) reciprocal, but CDs increased [32]. This can be attributed to the enormous progress in the socio-economic condition and changes in lifestyle in KSA [34]. In addition to The New Health Care Model as a part of the National Transformation Program, which launched in 2016 involves national screening programs focusing on diseases prevention and strengthening the primary health care system to screen risk or asymptomatic individuals, early diagnosis, and effective management [35].

In KSA, 2020, the top ten underlying causes of mortalities according to the ICD-10 in descending order were CVDs, infectious and parasitic diseases, injuries and poisoning, cancer, chronic respiratory disorders, (Road Transportation, and Traffic Accidents) RTTAs, Acute Respiratory Infections (ARIs), diabetes, Nervous System, and Sense organ disorders, maternal Conditions and neonatal causes, and Genitourinary diseases. That order changed from the top ten causes reported by the CDC, in 2018 ischemic heart disease, RTTA, stroke, chronic kidney disease, ARIs, Alzheimer’s disease, conflict and terror, cirrhosis, neonatal disorders, diabetes [36].

So currently, CVDs are considered the leading underlying cause of death in KSA [37,38] and both developing and developed countries [38]. The global agenda of NCDs have been expanded beyond the concept of four diseases to include mental health [39] in agreement with our finding that the mental, nervous, and sensory disorders became 678 (2.3%).

Regarding NCDs mortalities per region; as a cause per province-specific MR were Najran, Al Jouf, Al-Qounfodah, and Makkah in order by (9.1, 8.9, 8.3, 8.2) and the lowest was in The Eastern region 2.6 per 10,000 of the province population. This can be explained by disparities in the provided health services e.g., the quality, accessibility, and preparedness during the pandemic, in addition to the heterogeneity among provinces; difference in gender composition, social, economic, legal, cultural, and political factors [40].

NCDs mortalities per age; ASMR counts 26.4/100,000 children less than 5 year, then decreased to be <12/100,000 at the age groups from (five–<45 year), then increases from (45–<50 year) to be 29.1 and doubled at the age (50–<55 year) to be 62.3/100,000 and so on. Similar increasing trends. There were a significant difference in health status and NCDs mortalities between different age groups and sex can be attributed to the different levels of exposure and vulnerability to NCDs risk factors as reported by previous studies [41,42].

Regarding the pattern and trends of NCDs mortalities across months; (the number of mortality due to NCDs was 3019 at the end of January, then became 2146 at the end of April, to be 2894 by the 1st of July 2020. This can be explained by the effective national response to manage this crisis through comprehensive national isolation policies, e.g., lockdown, active surveillance models, lifestyle changes, Telemedicine online services 24/7, outpatient support (937 call center), the ability to refill their medication, with secure cross-platform applications (App) e.g., Sehaty, Asseafny, and Tetamen.

Regarding CDs mortality in KSA, during the first half of 2020, the most common five CDs (other than COVID-19) namely, Streptococcal infection, Varicella, Mycoplasma Pneumonia, Tuberculosis, and Hepatitis. While between 2003–2016, they were hepatitis B, dengue fever, measles, chickenpox, and brucellosis [43]. Inconsistent with the estimated GBD study about 2020, the top causes of disability-adjusted life-years are projected to be lower respiratory infections, tuberculosis, diarrheal diseases, and Human Immune Deficiency virus (HIV) [44].

Despite technological developments in diagnostics, treatment, and vaccination programs, various infectious diseases have re-emerged in KSA [45] especially during the Summer season and religious congregations in Holy sites [46]. So CDs surveillance and reporting systems should be assessed in terms of quality, efficiency, and effectiveness [47,48] to be controlled as a public health priority to prevent the spread of contagious diseases [49].

CDs mortalities per age ;we reported that the top three CDs causing infant mortality (IM) were Septicemias, Varicella, and (Acute Respiratory Infection) ARI, while in the same duration in China they were ARI, diarrhea, and septicemias [50]. WHO reported that most neonatal mortalities are caused by preterm birth, intrapartum-related complications (birth asphyxia or lack of breathing at birth), infections, and birth defects. This may be attributed to conditions and diseases associated with lack of quality care at birth or skilled care and treatment immediately after birth and in the first days of life [51].

Among children under 5 years, CDs remain the leading cause of mortality. In this study, 236 (14.3 %) of mortalities were due to CDs, higher than in China (19%) [49], while 744 (45.2%) were due to NCDs. While WHO reported preterm birth complications, birth asphyxia/trauma, pneumonia, congenital anomalies, diarrhea, and malaria as leading causes [51]. This is because CDs can be avoided through prevention and treatment, education, and immunization campaigns, to achieve the Sustainable Development Goals (SDGs) to end preventable death of newborns and children under 5 years of age and to end CDs by 2030 [52].

As regards the pattern and trends of CDs including COVID-19; the mortalities across months reveals a major dramatic increase from 617 (10.8%) at the end of January and 717 (18.5%) at the end of April to 1620 (26.2%) at 1 July. The forced lockdown process has unexpected effects on the diffusion of infectious diseases, due to blocking of the vaccination campaign, [43] decreasing flow to the Emergency Department (ED), due to a common fear of being infected during ED in-stay, and a marked reduction in accesses to the hospitals [53].

As regards, COVID-19 mortality Estimating COVID-19 mortality during a pandemic is challenged because it includes only the diagnosed cases, that depend on the test rate that depends on the of the Saudi MOH Protocol (SMOHP), and other factors such as prevention measures of public health, health response, etc... [54,55].

COVID-19 mortality per sex; male to female ratio for COVID-19 deaths was 2.76, that higher than what was reported by The European Centre for Disease Prevention and Control (ECDC) across the EU was 2.1, and what was reported by The U.K.’s Office for National Statistics (ONS) as it was 2.0. This variation may be due to biological factors (during viral infections, women mount great antiviral, inflammatory, and humoral immune responses, while men showed higher levels of lymphocytes, white blood cells, and neutrophils [56], and gender differences (the combined effect of differences in legal, social, cultural, political, and economic norms [57], similar results reported in Peru, China, Pakistan, and Indonesia similar social, and cultural factors. In contrast, countries such as Canada and many European countries [58].

COVID-19 mortality per age group; ASMR/100 000 population of the same age group to be less than 2 in the population (<35 year), and progressively increased after the age group (55–<60 year) to be 17.7, then doubled at the age group (60–<65 year) and so on till reaching the highest 74.9 at the age group 80 year and more. So people >65 year have a multi-fold mortality risk than younger, consistent with previous reports [59,60,61,62] because aging deteriorates innate immunity thus increases the severity of infection, and risk of mortality. Elderly people are likely to be affected more due to the restrictive effect of COVID-19 related lifestyle changes, access to healthcare; which is expected to worsen the underlying medical conditions such as diabetes, hypertension, obesity, and dyslipidemia [61,62].

COVID-19 mortality per regions; There are variations in the COVID-19 MR among the elderly in various geographical regions of the world, and the percentage of deaths in the total closed cases is different across the provinces, being 1.60% in Makkah Province, 0.24% in Riyadh Province, 0.28% in Eastern Province, 0.57% in Al Madinah Province and 0.33% in Gazan Province [63] nearly similar to our statistics.

The infectious disease mortalities were 5005 (17.1%) due to the COVID-19 deaths. While 2773 (9.4%) of mortalities were caused by injuries (unintentional and intentional), which stated by WHO to become more prevalent and were projected to show rates equal to mortality from infectious diseases by 2020. The SAMOH in collaboration with the WHO has emphasized improving the health situation by focusing on five key health issues; namely, (1) health security and disease prevention, (2) control of communicable diseases, (3) control of non-communicable diseases, (4) accidents and injuries, and (5) strengthening of health systems, on a priority basis [53,64]

Regarding (Road Transportation, and Traffic Accidents) RTTAs mortalities WHO considered KSA to have the world’s highest mortalities from RTTAs, which was the main cause of death in adult males aged 16 to 36 year [65]. This evidence supports that the SA action plan focused mainly on the prevention of RTTIs rather than RTTAs, including seat belt laws, SAHER system, emergency medical services, and the role of the police in the documentation of RTAs [66]. While the international recommendations focused on developing an institutional framework, safer roads and vehicles, proper surveillance, and post-crash care [67].

As regards RTTA mortalities per region, KSA is a vast country of 2,149,690 km^2^ [56]. The top regions regarding the occurrence of RTTAs mortalities were Riyadh, Makkah, Jeddah, and Asser was the highest 23.3%, 9.3%, 7.8%, and 7.4% and the lowest was in Al Jouf (0.4%). This can be explained by the highest population density (per km^2^) as it is 101 in Jazan, and 38 in Makkah, and the lowest of 2.8 in Najran, and 3.6 in Al Jouf [65]. While in previous studies, the top regions were Riyadh, Jeddah, Makkah, AL-Madinah al Monawah, and Al-Qassem [68].

RTTAs mortalities trends across months; the RTTAs mortality reduced from 4.3% in February to 2.4% in April (Ramadan) then gradually increased with gradual reopen. Due to the lockdown, restricted travel, and recreation activities [69]. The months of RTTAs occurrence reported in few studies to be at the lowest rate 6.4% during February, while in Al Qassim, it was 5.8% in March [(16] inconsistent with our finding. Another study [70] reported that Ramadan (April) is the most common month for occurrence of RTTAs.

After the announcement of COVID-19 as a pandemic. This epidemiological trend has distressed most national healthcare services in developed countries with some extremely overwhelmed. Resulting in potentially, unexpected worse health outcomes due to the combination of frailty, aging, which affect NCDs, and increases the overall mortality [4,5]. Therefore, it is time to analyze the effects caused by the long lockdown period.

COVID-19 is a major global crisis [3], and its death certificate has a considerable role in the efforts to control the virus spread so that many countries have taken steps to ensure its accurate reporting [71]. It has resulted in lockdown in many parts worldwide. Globally, till date 04 July 2020 (a total of 11,209,025 confirmed cases with 529,471 deaths and 6,356,170 completely recovered) [72].

Over months from the 30th of April till the 1st of July 2020, Although the strict preventive measurements and lockdown, but the pandemic morbidities and mortalities progressively increased thus altering the MR, so that the reported estimation of COVID-19 fatality rate in KSA had increased from 2% to 3.4% [73], inconsistent with most countries in which MR declined (except Italy, Mexico, and Germany), this happens due to disproportionation between the recovery rate and death rate [1].

The pattern of the outbreak differs considerably from the pattern experienced in Europe [43]. Similar patterns were seen in earlier outbreaks from the Coronavirus family, perhaps due to the similarities in the origin and the disease development. These patterns present another view of a neglected aspect of healthcare response, management of chronic disease, also an opportunity for health systems improvement, especially in emergency preparedness.

The sex and nationalities had a statistically significant relationship between the main underlying causes of death (*p* < 0.05). This is because the differences in health-related behavior/lifestyle play an important role in explaining this significant relationship [74].

### 4.1. Limitations

There are many limitations; (1) There was no baseline comparison to other years; (2) Limitations of death reporting data; e.g., delayed reporting. Death reporting is affected by many considerations; The deaths that occur outside the health settings may be unattended, asymptomatic patients with mild clinical symptoms may be misdiagnosed or be ill-defined i.e., under-reporting of certain causes, and patients dying with infectious diseases are deemed to be dying from the disease.

### 4.2. Strength

This study is honest and transparent reporting about the mortality during the first wave of the COVID-19 pandemic. It is a baseline comparison comprehensive relevant national-based mortality data represents mortalities in KSA at the whole age groups, sex, provinces, causes, and nationalities, so the results can be generalized; meanwhile, the majority of studies were based on aggregate mortality data to serve limit group age, sex, and sex even the (GBDS) [75]. The COVID-19 mortalities were verified and confirmed through the Saudi National COVID-19 committee.

## 5. Conclusions

During the COVID-19 pandemic on the FIRST half of 2020 in KSA; NCDs mainly CVDs were the leading underlying causes of death; Male gender and old age > 55 year was a risk factor for worse outcomes and death in COVID-19 cases. The top six underlying causes of mortalities from CDs (other than COVID-19) were Streptococcal, varicella, Plaque, Mycobacterium Pneumonia, Hantavirus, and Cholera. Among children under 5 years, CDs remain the main underlying cause of mortality; The top regions regarding the occurrence of RTTAs mortalities were Riyadh, Makkah, Jeddah, and Asser; The lockdown was associated with a manifest reduction in the total number of deaths due to NCDs and RTTAs but increases the CDs mortalities. Sex and nationality were significantly associated with the main underlying causes of mortalities.

## 6. Recommendations

Many recommendations at different levels to strengthen the medical certification of death to be helpful guide for logistic planning; (1) at the level of documenting the cause of death in death certificate; by providing specialized physician, training programs on nosology as only one cause of death in death certificate except in complex causes a systematic classification is required, and improving the services to detect the causes of deaths. (2) At the level of coding; through providing well-trained persons in coding known as a nosologist because of the complexity of cause of death coding leads to specialized occupation/certification as a qualified nosologist. (3) At the registration level; to be done only by trained qualified persons with continuous training and evaluation; (4) Increasing the number of mortality reporting health settings considering the population density and geographical accessibility. (5) Using a national specialized Soft Ware Program for online register with special requirement (obligate identity document (ID), and rejecting null causes or mismatched underlying and main causes); (6) At the future policy and services; using this up-to-date and reliable population-based statistics of mortality for needs assessments on national reprehensive base e.g., budget allocation and guided effective national programs based on the national burden of diseases e.g., Central Nervous System (CNS) programs. (7) Conducting National Researches to understand the patterns, trends, and main determinants to come up with evidence-based interventions and strengthen health system response.

## 7. Ethical Considerations

The collected data were managed privately and the identity was anonymous. The ethical approval was taken from the ethical committee of the research center at King Fahad Medical City IRB Log Number: 20-551E.

## Figures and Tables

**Figure 1 ijerph-18-12647-f001:**
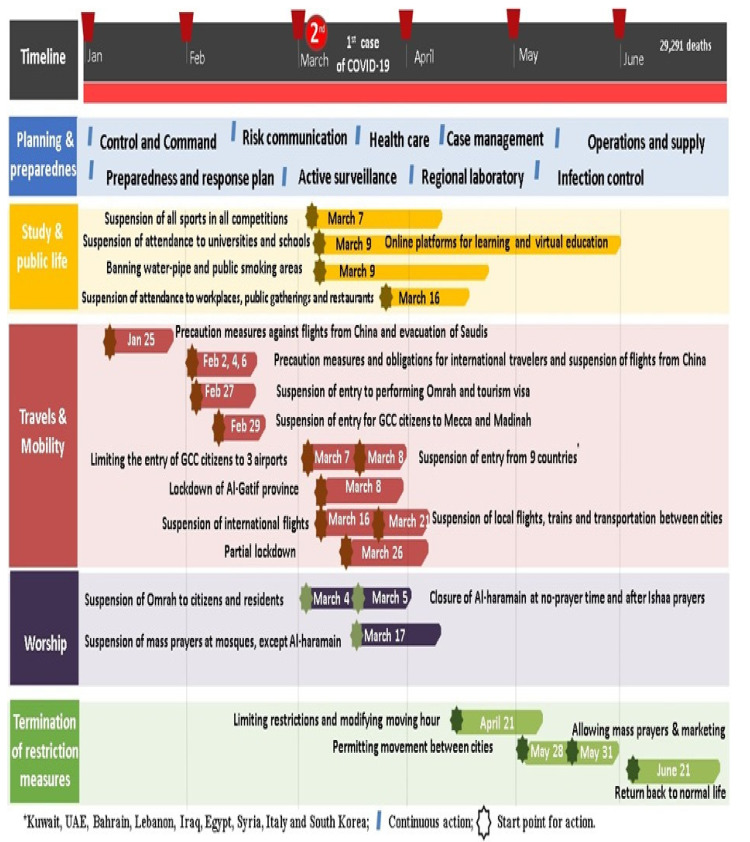
The flow of governmental restriction measures against COVID-19 in KSA, 2020.

**Figure 2 ijerph-18-12647-f002:**
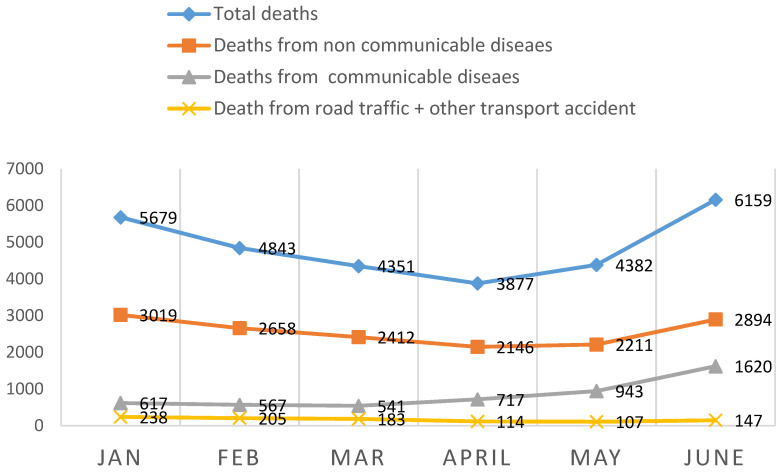
The pattern of the Mortality from the 1 January–30 June 2020 in KSA across months considering the governmental restriction procedures.

**Table 1 ijerph-18-12647-t001:** Causes of deaths according to The International Classification of Diseases (ICD-10 codes) in KSA from first of January until 30 June 2020.

Diseases and Injury Categories and ICD-10 Codes	Underlying Causes F (%)	F %
**I-Communicable diseases (CDs), maternal, and neonatal conditions**		
A-Infectious, parasitic diseases	2438 (8.3)	5141 (17.6)
B-Acute Respiratory Infections	864 (2.9)	
C-Maternal Conditions	177 (0.6)	
D-Neonatal Causes	351 (1.2)	
E-Nutritional deficiencies	13 (0.0)	
COVID-19 as the cause of disease classified to other chapters	1298 (4.4)	
**II-Non Communicable diseases (NCDs)**		
F-Malignant + other neoplasms	1199 (4.1)	11,862 (40.5)
G-Diabetes Mellitus	658 (2.2)	
H-Endocrine and metabolic disorders	232 (0.8)	
I-Mental disorders	38 (0.1)	
J-Nervous System, Sense organ disorders	640 (2.2)	
K-Cardiovascular Diseases (CVD)	6864 (23.4)	
L-Chronic Respiratory diseases	1153 (3.9)	
M-Diseases of the digestive system + Oral health	255 (0.9)	
N-Genitourinary diseases	483 (1.6)	
O-Skin diseases	29 (0.1)	
P-Musculoskeletal disease	96 (0.3)	
Q-Congenital anomalies	189 (0.6)	
R-Ill -defined conditions	26 (0.1)	
**III-Injuries**		
**S-Unintentional injuries T = 2773 (9.4)**		10,213 (34.9)
Road traffic+ other transport accidents	994 (3.4)	
Unintentional injuries	184 (0.6)	
Injury, poisoning, and certain other consequences of external causes	1421 (4.9)	
**T-Intentional injuries**		
	174 (0.6)	
**U-Codes allocated or redistributed to other codes**		
**T = 3071 (10.5)**		
Ill-defined malignant neoplasm + Unspecified uterine cancer	146 (0.5)	
Ill-defined unintended accidents	48 (0.2)	
Falls, Burn, Poisoning, Drowning, Suffocation	22 (0.1)	
Ill-defined cardiovascular conditions +heart failure	2768 (9.5)	
Ill-defined gastrointestinal conditions	87 (0.3)	
**Ill-defined conditions; T = 4369 (14.9%)**		
Senility	192 (0.7)	
Sudden death	399 (1.4)	
Unknown, nonspecific cause of morbidity	114 (0.4)	
Instantaneous death	469 (1.6)	
Death < 24 h	253 (0.9)	
Unattended death	1083 (3.7)	
Other ill-defined cause of Mortality	949 (3.2)	
Other CVD conditions (abnormal rhythm, blood pressure, and strokes)	471 (1.6)	
Others	439 (1.6)	
**IV-Codes of special Conditions**		
SARS	15 (0.1)	473 (1.6)
Emergency use of U07.1 [COVID-19, virus identified]	390 (1.3)	
Emergency use of U07.2 [COVID-19, virus not identified]	51 (0.2)	
Firearm +Horse race +hyperactivity	17 (0.1)	
**V-Factors that influence the health status**	117 (0.4)	117 (0.4)
**Null**	1485 (5.0)	1485 (5.0)
**Total**	29,291	100.0

**Table 2 ijerph-18-12647-t002:** Main underlying causes of mortality from 1 January–30 June 2020 in KSA according to ICD-10 categorization and WHO classification.

WHO Classification of Main Causes of Deaths	The First Half of 2020 Underlying Causes of Mortality F(%)
**Communicable, maternal, perinatal, and nutritional conditions**	5546 (18.9)
(including SARS, and COVID-19 confirmed cases by special codes)	
**Diabetes Mellitus**	658 (2.2)
**Chronic Respiratory diseases**	1053 (3.9)
**Cardio-vascular diseases**	
{ICD-10, heart Failure-ill-defined and unspecific Cardiovascular conditions+ heart failure + abnormal heart rhythm+ abnormal blood pressure}	10,103 (34.5)
**Cancer**	
(Malignant+ other neoplasm + Ill-defined malignant neoplasm + Unspecified uterine cancer)	1345 (4.6)
**Other Non-communicable diseases (NCDs)**	2181 (7.4)
**Injuries**	
Other Injuries Road traffic+ other transport accidents	1866 (6.4)
	994 (3.4)
	T = 2860 (9.8%)
Ill-defined injury conditions /unknown/risk factors	4060 (13.9)
Null	1485 (5.0)
No of deaths	29,291
Total no of the population (1440 h, 2019 G)	34,218,168

**Table 3 ijerph-18-12647-t003:** Distribution of the main underlying causes of mortalities by age groups.

Age Groups (y)	Total Number (f)	Proportional Mortality (%)	NCDs Mortality F (%)	NCDs-Age Specific ASMR *	CDs ** Mortality F (%)	CDs-ASMR *	COVID-19 Mortality F (%)	COVID-19 ASMR *
Abortion				------		--------		-----
<16 w	47	0.2	0		0 (0.0)		0 (0.0)	
>16 w	236	0.8	23 (0.1)		3 (0.1)		0 (0.0)	
1 m	645	2.2	314 (2.0)	744 26.4	101 (3.04)	236 8.4	1 (0.1)	3 0.01
1 m–12 m	225	0.9	69 (0.4)		14 (0.4)		----	
1–4	776	2.6	361 (2.3)		121 (3.6)		2 (0.1)	
5–9	225	0.8	115 (0.7)	3.8	19 (8.4)	0.64	0	---
10–14	227	0.8	97 (0.6)	3.7	21 (0.6)	0.8	0	----
15–19	514	1.8	163 (1.1)	6.9	34 (1.0)	1.4	3 (0.2)	0.13
20–24	850	2.9	216 (1.4)	8.2	49 (1.5)	1.9	9 (0.5)	0.34
25–29	1049	3.6	343 (2.2)	10.5	75 (2.3)	2.3	20 (1.2)	0.61
30–34	1107	3.8	424 (2.7)	12.9	78 (2.4)	2.4	49 (2.9)	1.48
35–39	1119	3.8	481 (3.1)	12.97	82 (2.5)	2.2	84 (4.9)	2.27
40–44	1361	4.6	666 (4.3)	20.1	96 (2.9)	2.9	146 (8.6)	4.49
45–49	1504	5.1	723 (4.7)	29.3	131 (3.9)	5.3	150 (8.8)	6.9
50–54	2019	6.9	1067 (6.9)	62.23	173 (5.2)	10.1	245 (14.5)	14.28
55–59	2177	7.4	1195 (7.7)	100.9	210 (6.3)	17.7	209 (12.4)	17.66
60–64	2689	9.2	1505 (9.8)	193.2	286 (8.6)	36.7	229 (13.6)	29.39
65–69	2222	7.6	1296 (8.3)	298.7	254 (7.7)	58.6	177 (10.5)	40.87
70–74	2225	7.6	1364 (8.8)	473.5	292 (8.8)	101.4	108 (6.4)	37.5
75–79	2375	8.1	1451 (9.4)	839.5	353 (10.6)	204.2	103 (6.1)	59.6
80–84	2148	7.3	1338 (8.7)	3267 1597.7	333 (10.04)	1104 529.5	69 (4.1)	153 74.8
85–89	1692	5.8	1033 (6.6)		267 (8.0)		48 (2.8)	
90–94	1054	3.6	629 (4.1)		176 (5.3)		25 (1.5)	
95–99	447	1.5	77 (0.4)		273 (8.2)		8 (0.5)	
100–104	208	0.7	128 (0.8)		35 (1.1)		2 (0.1)	
105–109	82	0.3	41 (0.3)		16 (0.5)		1 (0.1)	
110 or more	38	0.1	21 (0.1)		4 (0.1)		----	
Total	29,291	100.0	15,430		3317		1688	

W = week m = month * Age-Specific Mortality Rates are * by 100,000. ** CDs communicable diseases other than COVID-19.

**Table 4 ijerph-18-12647-t004:** Distribution of the main underlying causes of mortality according to the health Provinces in KSA.

Health Province	No of Deaths	Proportional Mortality (PM)	Province Specific MR% * 10,000	NCDs Proportional Mortality F (%)	NCDs-Province Specific MR * 10,000	RTTA Proportional Mortality F (%)	RTTA Region-Specific MR * 100,000
Makkah	4270	14.6	17.4	1936 (12.6)	8.2	92 (9.3)	3.7
Riyadh	5271	18.0	6.1	2021 (13.2)	5.5	232 (23.3)	2.7
Jeddah	3616	12.3	7.3	2620 (17.1)	5.3	78 (7.8)	1.6
Al Madinah-Al Monawarah	2355	8.0	10.5	976 (6.4)	4.3	57 (5.7)	2.5
Aseer	1856	6.2	9.76	1010 (6.6)	5.3	74 (7.4)	3.9
Jazan	2151	7.3	13.1	1257 (8.8)	7.6	57 (5.7)	3.4
Al-Qassem	1280	4.4	8.6	573 (3.7)	3.9	18 (1.8)	1.2
Al Taif	1290	4.4	9.48	683 (4.5)	5.2	44 (4.4)	3.3
Eastern Region	1760	6.0	5.17	896 (5.8)	2.6	66 (6.6)	1.9
Tabouk	714	2.4	7.5	314 (2.0)	3.3	52 (5.2)	5.4
Hail	673	2.3	9.2		---	65 (6.5)	8.8
Al-Qunfudah	386	1.3	11.9	270 (1.8)	8.3	10 (1.0)	3.1
Najran	552	1.9	9.07	308 (2.0)	9.1	30 (3.0)	4.9
Al-Jouf	471	1.6	13.3	317 (1.8)	8.9	4 (0.4)	1.1
Al-Ahsa	784	2.7	6.1	553 (70.5)	4.3	27 (2.7)	2.1
Al-Bahaa	486	1.7	9.8	309 (2.0)	6.2	2 (0.2)	0.4
Bisha	399	1.4	9.8	290 (1.9)	7.1	6 (0.6)	1.5
Northern Border	435	1.5	11.3	270 (1.8)	7.0	6 (0.6)	1.6
Hafr -Al Baten	385	1.3	8.2	224 (1.5)	4.7	24 (2.4)	5.1
Al-Qurayyat	157	0.5	8.8	92 (0.6)	5.2	10 (1.0	5.6

* Saudi Statistical book, 2019 (used for the dominator).

**Table 5 ijerph-18-12647-t005:** Distribution of the main underlying causes of mortalities according to sex.

	Sex-Specific Mortality Rate (MR) * 100,000	Proportional NCDs Mortality Per Sex F (%)	NCDs Sex-Specific MR(SSMR) * 100,000	CDs Proportional MR Per Sex	CDs SSMR * 100,000	Proportional COVID-19 Mortality Per Sex F(%)	COVID-19 SSMR * 100,000
**Sex**							
**Males (19,212)**	97.9	9424 (49.1)	47.7	1899 (9.8)	9.8	1361 (7.1)	6.9
**Females(10,079)**	95.8	5916 (58.7)	40.8	1418 (14.1)	100.1	327 (3.2)	2.5

* Saudi Statistical book, 2019.

**Table 6 ijerph-18-12647-t006:** The relationship between the main underlying causes of deaths and sex and nationality in KSA.

	Sex		Nationality	
	Males N = 19,212 F (%)	Females N = 10,079 F (%)	Saudi N = 18,110 F (%)	Non-Saudi N= 11,181 F (%)
**Null (N = 1485)**	938 (4.9)	547 (5.4)	793 (4.4)	692 (6.2)
**RTTA (N = 994)**	867 (4.5)	127 (1.3)	640 (3.5)	354 (3.2)
**CDs ***(N = 3317)**	1899 (9.9)	1418 (14.0)	2409 (13.3)	908 (8.1))
**NCDs (N = 15,340)**	9424 (49.1)	5916 (58.7)	10,394 (57.4)	4946 (44.2)
**COVID-19 (N = 1688)**	1361 (7.1)	327 (3.2)	463 (2.6)	1225 (11.0)
**Other causes ****	4723 (24.7)	1744 (17.3)	3411 (18.8)	3056 (27.3)
** *p* **	0.01 *		0.00 *	

* *p* < 0.05—there was a significant difference; ** includes (Other injuries, ill-defined conditions of injury, factors related to the health service, maternal and neonatal causes); *** CDs—Communicable diseases other than COVID-19.

## Data Availability

The datasets used during the current study are available from the corresponding author or the principle investigator on reasonable request.

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
