# Peer review of "Determinants of the Underlying Causes of Mortality during the First Wave of COVID-19 Pandemic in Saudi Arabia: A Descriptive National Study"

_ijerph, 2021, doi:10.3390/ijerph182312647_

Round 1

Reviewer 1 Report

In their descriptive national study (Kingdom of Saudi Arabia), the authors have reported data on the mortality during the first wave of the COVID-19 pandemic. This retrospective analysis is well conducted and discussed, with tables and graphs that clarify the big amount of data recorded. I have also appreciated Figure ,1 which describes the flow of governmental restriction measures against COVID-19 and the short paragraph at the end of the paper describing Recommendations about medical certification of death.

It is a great pleasure to review this well-prepared manuscript.

Author Response

Thank you very much 

Reviewer 2 Report

Dear Authors,

I have the following comments on your paper "Determinants of the Underlying Causes of Mortality during the First wave of COVID-19 Pandemic in Saudi Arabia. Descriptive National Study."

  • Line 23: please use p<0.05 for significant differences; p=0.00 is invalid.
  • please provide references in the order they appear in the text (e.g., line 34 shows reference number 83?
  • please use uniformly either SA or KSA throughout the text
  • please check and correct spaces, punctuation marks, brackets, large and lower case, etc. throughout the text; 
  • please do to use abbreviations unless explaining them (e.g. line 102, "T.B"
  • please explain which hepatitis in included in line 102
  • please explain "RTTA"
  • please discuss potential causes for regional differences of data

Best wishes

Author Response

  • Line 23: please use p<0.05 for significant differences; p=0.00 is invalid. (DONE )
  • please provide references in the order they appear in the text (e.g., line 34 shows reference number 83? . (DONE )
  • please use uniformly either SA or KSA throughout the text(DONE )
  • please check and correct spaces, punctuation marks, brackets, large and lower case, etc. throughout the text;  (Done )
  • please do to use abbreviations unless explaining them (e.g. line 102, "T.B"(DONE )
  • please explain which hepatitis in included in line 102--- according to ICD-10 as in refe (20)
  • please explain "RTTA""(DONE )  according to ICD-10 as in refe (20)

please discuss potential causes for regional differences of data ----- at lines ( 186-189) 

Reviewer 3 Report

This article is quite timely as globally many are looking for direction as to how to prepare for the next event. KSA is a good case study of how many countries are structured, with large immigrant populations who are mostly transitory and also aging. This article validates what many others are findings in their studies that those with pre-existing conditions, aged populations, and other underlying conditions are more likely to have adverse and long-term effects. Line 30-32 needs to be restructured. The way it written reads poorly. Line 59 remove "if existed." It adds confusion to the meaning of the sentence Line 64-66 needs to be rewritten. Suggest starting the sentence "The study attempted to determine the demographic characteristic, patterns, determinants, and the main causes of death......" Line 101, Pneumonia needs to be capital Results Section is brief but provides sufficient information. Line 148, It could affect the analysis and may bias the results (not biased). Lines 184 to 188 does not make sense to me. Thus, I'm not sure of the point you are making in the paragraph. Line 197 to 202. After reading it a few times, I understand that you are trying to convey that with the added precaution of lockdown, NCD related additional deaths from COVID-19 were averted, but it doesn't read clearly. Please rewrite. Line 216 to 218. It's good that you are exploring infant mortality rates, but why compare to China? There needs to be more background, if you are using comparisons. The points you are making in the paragraph makes sense, but you don't have sufficient set-up. Line 223 to 230, same comment as above. Why China? Justification for comparing it KSA? Climate? Culture? Industry? GDP? Line 242 to 243, you bring in the European Union (EU). Not sure of the context, similar to my argument above, regarding China. Need more context. Conclusion The conclusion section needs to be expanded. You brought up a lot of points in the discussion and they need to be addressed here. You're only addressing gender. Recommendation I'm glad you added a recommendation (Best Practice) section to your report; but, it needs major work. Your discussion section was the strongest part of your paper and some of those points need to brought up here, with suggested improvements. I don't see any of that here.

Author Response

  • 59 remove "if existed." Done
  • It adds confusion to the meaning of the sentence Line 64-66 needs to be rewritten. Suggest starting the sentence "The study attempted to determine the demographic characteristic, patterns, determinants, and the main causes of death......",

The study attempted to determine the demographic characteristic, patterns, determinants, and undelying causes of mortality during the 1st half of 2020 during the first wave of COVID-19 pandemic in KSA.

  • Line 101 Pneumonia needs to be capita (done )

We founded that the main CDs underlying causes of mortalities  based on ICD-10 (20) were 3317(11.3%), out of them 167(5.0) Streptococcal infection, 95(2.9%) Varicella 90(2.7%) Plaque, 78(2.4%) Mycoplasma Pneumoniae, 70(2.1%) Hanta virus, 66(1.9%) Cholera, 56(1. 7%) Tuberculosis (TB), 42(1.3%) Hepatitis, 27 (0.008%), Candidiasis, 18(0.05%) Meningococcal infection, 13(0.004%) Human Immune Deficiency virus, 11(0.003%) Typhoid and Salmonella, and 10(0.003%) Hemorrhagic fever .

  • Results Section is brief but provides sufficient information. Thank you
  •  
  • Line 148, It could affect the analysis and may bias the results (not biased)---- removed
  • Lines 184 to 188 does not make sense to me. Thus, I am not sure of the point you are making in the paragraph.

       The global agenda of NCDs has been expanded beyond the concept of four diseases to include mental health (39) in      agreement with our finding that the mental, nervous, and sensory disorders became 678 (2.3%).{ means the importance of considering mental health disorders beside CVDs, DM, Cancer, ACCIDENTS )

  • Line 197 to 202. After reading it a few times, I understand that you are trying to convey that with the added precaution of lockdown, NCD related additional deaths from COVID-19 were averted, but it doesn't read clearly. Please rewrite.

As regards the pattern and trends of NCDs mortalities across months;  (the number of mortality due to NCDs  was 3019 at the end of January , then became  2146  at the end of April, to be 2894 by the 1st of July 2020. This can be explained by the effective national 

  • Line 216 to 218. It's good that you are exploring infant mortality rates, but why compare to China? There needs to be more background, if you are using comparisons the points you are making in the paragraph makes sense, but you do not have sufficient set-up. Line 223 to 230, same comment as above. Why China? Justification for comparing it KSA? Climate? Culture? Industry? GDP? Line 242 to 243, you bring in the European Union (EU). Not sure of the context, similar to my argument above, regarding China.------- there wasn’t any other published studies regards mortality during this period in COVID-1p pandemic except in CHINA , and Europe
  • . Conclusion The conclusion section needs to be expanded. You brought up a lot of points in the discussion and they need to be addressed here. You are only addressing geder ------- expanded

During the COVID-19 pandemic on the 1ST  half of 2020 in KSA;  NCDs mainly CVDs were the leading underlying causes of death; Male gender and old age >55y was a risk factor for worse outcomes and death in COVID-19 cases. The top sex undelring causes of mortalities from CDs (other than COVID-19) were Streptococcal, varicella, Plaque, Mycobacterium Pneumonia, Hantavirus, and Cholera; Among children under 5 years, CDs remain the mainunderlying cause of mortality; The top regions regarding the occurrence of RTTAs mortalities were Riyadh, Makkah, Jeddah, and Asser ;The lockdown was  associated with a manifest reduction in the total number of deaths due to NCDs and RTTAs but increases the CDs mortalities. Sex and nationality were significantly associated with the main underlying  causes of mortalities

  • Recommendation I'm glad you added a recommendation (Best Practice) section to your report; but, it needs major work Your discussion section was the strongest part of your paper and some of those points need to brought up here, with suggested improvements. I don't see any of that here-------

Many recommendations at different levels  to strengthen the medical certification of death to be helpful guide for logistic planning ;1) at the level of documenting the cause of death in death certificate ; by providing specialized physician, training programs on nosology   as only one cause of death in death certificate except in complex causes a systematic classification is required, and improving the services to detect the causes of deaths .2) At  the level of  coding ; through providing well-trained persons in coding known as a nosologist because of  the  complexity of  cause of death coding leads to  specialized occupation/ certification as a qualified nosologist.3)At the registration level ;to be done only by trained qualified persons with  continuous training and evaluation;  4)Increasing the number of mortality reporting health settings considering the population density and geographical accessibility.5)  Using a national specialized Soft Ware Program for online register with special requirement (obligate ID, and Rejecting null causes or mismatched underlying and main causes); 6 ) At the future policy and  services ; using  this up-to-date and reliable population-based statistics of mortality for  needs assessments on national reprehensive base  e.g. budget allocation and guided effective national programs based on the national burden of diseases e.g. CNS programs .7)Conducting   National Researches to understand the patterns, trends, and main determinants to come up with evidence-based interventions and strengthen health system response.

Round 2

Reviewer 2 Report

Dear Authors,

thank you for the corrections in the manuscript, I will recommend accept in present form.

Reviewer 3 Report

Line 32, Kingdom is misspelled 

Line 341: Six not sex